

# Inorganic Carbon Cycling and Biogeochemical Processes in an Arctic Inland Sea (Hudson Bay)

William J. Burt[1], Helmuth Thomas[2], Lisa A. Miller[3], Mats A. Granskog[4], Tim N. Papakyriakou[5], Leah Pengelly[2]

[1]Department of Earth, Ocean and Atmospheric Sciences, University of British Columbia, Vancouver, British Columbia, Canada
[2]Department of Oceanography, Dalhousie University, Halifax, Nova Scotia, Canada
[3]Institute of Ocean Sciences, Fisheries and Oceans Canada, Sidney, British Columbia, Canada
[4]Norwegian Polar Institute, Fram Centre, NO-9296 Tromsø, Norway
[5]Centre for Earth Observation Science, University of Manitoba, Winnipeg, Manitoba, Canada

*Correspondence to:* William J. Burt (wburt@eos.ubc.ca)

**Abstract.** The distributions of carbonate system parameters in Hudson Bay, which not only receives nearly one third of Canada's river discharge, but which is also subject to annual cycles of sea-ice formation and melt, indicate that the timing and magnitude of freshwater inputs play an important role in carbon biogeochemistry and acidification in this unique Arctic ecosystem. This study uses basin-wide measurements of dissolved inorganic carbon (DIC) and total alkalinity (TA), as well as stable isotope tracers ($\delta^{18}O$ and $\delta^{13}C_{DIC}$), to provide a detailed assessment of carbon cycling processes within the bay. Surface distributions of carbonate parameters reveal the particular importance of freshwater inputs in the southern portion of the bay. Based on TA, we surmise that the deep waters in the Hudson Bay are largely of Pacific origin. Riverine TA end-members vary significantly both regionally and with small changes in near-surface depths, highlighting the importance of careful surface water sampling in highly stratified waters. In an along-shore transect, large increases in subsurface DIC are accompanied by equivalent decreases in $\delta^{13}C_{DIC}$ with no discernable change in TA, indicating a respiratory DIC production on the order of 100 μmol kg$^{-1}$ during deep water circulation around the bay.

**Keywords:** carbon cycling, carbonate system, freshwater, stable isotopes, Hudson Bay System.

## 1. Introduction

The Arctic Ocean is particularly vulnerable to "Ocean Acidification" (defined as the combined results of decreasing pH and increasing calcium carbonate solubility), because of a combination of high $CO_2$ solubility and low buffer capacity in cold waters, naturally high $CO_2$ concentrations in inflowing Pacific water, and dilution from sea-ice melt and river waters (AMAP, 2013). Therefore, regions of the Arctic Ocean are predicted to be among the first to experience the damaging effects of ocean acidification (Orr et al., 2005; Fabry et al., 2009). Freshwater input can directly reduce the buffering capacity of seawater, and dilute carbonate ions, thereby decreasing the saturation states of calcite ($\Omega_{Ca}$) and aragonite ($\Omega_{Ar}$), minerals that many important marine species require to form their shells (e.g., Chierici and Fransson, 2009; Yamamoto-Kawai et al., 2009).

The Hudson Bay system receives nearly one third of Canada's river discharge, and Hudson Bay itself goes from complete ice-cover in winter to open water in summer, culminating in an annual freshwater yield from river runoff and sea-ice melt that is more than double that of the Arctic Ocean (Granskog et al., 2011). Organic matter respiration releases dissolved inorganic



carbon (DIC) and consumes total alkalinity (TA), which decreases the $\Omega_{Ca}$ and $\Omega_{Ar}$ in deep waters. In addition to the organic matter generated in the euphotic zone by primary producers, local riverine inputs of organic matter are injected into deep waters during sea-ice formation (Mundy et al., 2010; Granskog et al., 2011). Furthermore, at the mouth of Hudson Bay shallow sills restrict the exchange of deep water between Hudson Bay and the relatively well-ventilated waters from the adjacent Hudson

Strait or Foxe Strait (Granskog et al., 2011). Given these characteristics, Hudson Bay may be particularly vulnerable to ocean acidification. In support of this, Azetsu-Scott et al. (2014) recently provided the first basin-wide overview of the Hudson Bay carbonate system in fall 2005, reporting that surface waters in south-eastern Hudson Bay, where freshwater inputs are highest, were undersaturated with respect to aragonite ($\Omega_{Ar} < 1$), as were up to two-thirds of Hudson Bay bottom waters.

With air temperatures rising, sea-ice coverage declining, and increasing river diversion for hydroelectricity generation,

conditions in the Arctic, and Hudson Bay in particular, are changing much more rapidly than in much of the world's oceans. The Hudson Bay system is one of the richest eco-regions for marine mammals in the world and is critical for both resident and migratory species (Wilkinson et al., 2009). Regime shifts in this ecosystem could therefore result in cascading effects impacting multiple organisms and coastal communities. The 2005 fall survey by Azetsu-Scott et al. (2014) provided an initial baseline for understanding of the state of the $CO_2$ system in Hudson Bay, yet significant uncertainties remain regarding the biogeochemical

processes responsible for the observed state of the marine carbonate system and how the system and its controlling processes have changed, or may continue to change, over time.

Here, we present recent (July 2010) seawater measurements of the marine carbonate system along with stable isotope ratios of oxygen in seawater ($\delta^{18}O$) and carbon ($\delta^{13}C_{DIC}$) across Hudson Bay. Surface distributions highlight the impact of different freshwater inputs on the carbonate system. Relationships of DIC and TA with salinity in deep waters provide insight into the

origin of waters to Hudson Bay, while similar relationships in shallow water describe key sources and sinks of carbon. We also evaluate the evolution of water mass properties during transit around Hudson Bay and the importance of precise sampling techniques when working in highly stratified waters.

## 2. The Hudson Bay System

The Hudson Bay system, which includes James Bay to the south, and both Foxe and Hudson Straits to the north, is shown in

Figure 1. Hudson Bay, itself, is the largest inland sea in North America (Martini, 1986), covering an area of 841,000 km$^2$ (Kuzyk et al., 2009). Hudson Bay is relatively shallow, with an average depth of 125 m and a maximum depth of 250 m, while areas of Foxe Strait and Hudson Strait reach depths of 400 m (Prinsenberg, 1987; Saucier et al., 2004). Water enters the Hudson Bay system from the Canadian Archipelago via Fury and Hecla Strait and Foxe Basin (predominantly waters of Pacific origin), and from the Labrador Sea (and ultimately, the North Atlantic) via Hudson Strait (Ingram and Prinsenberg, 1998). Hudson Bay is

connected to the Foxe and Hudson Straits via four main channels. The narrow channel west of Southampton Island is only 50 m deep but is considered an important source of water to the northwest corner of the bay (Prinsenberg, 1986). The majority of water exchange occurs via three channels located between southwest Southampton Island and the east coast of Hudson Bay. The outer channels are only 130 m deep, thus exchange of deep waters is likely limited to the central channel, which contains a sill at 185 m depth (Figure 1, black star).



Within Hudson Bay, the circulation is generally cyclonic (counterclockwise), with a mean summertime current velocity of 0.05 m s$^{-1}$ (Martini, 1986; Prinsenberg, 1986). During transit, coastal surface waters are substantially modified by river inputs, which total approximately 760 km$^3$ yr$^{-1}$ (Déry et al., 2011). The vast majority of this input enters via James Bay (47 % of the total) and from rivers along the southern coast (from the Churchill to the Great Whale, see Figure 1) that drain directly into

Hudson Bay (32 % of the total input). The peak river input (~2 km$^3$ day$^{-1}$) occurs in May, while the minimum (~1 km$^3$ day$^{-1}$) occurs in March (Déry et al., 2011). Freshwater is also added to surface waters via sea-ice melt (SIM). Hudson Bay is completely ice covered for 8-9 months of the year and becomes completely ice free in the summer. Sea-ice can reach a maximum thickness of about 1 m in James Bay, 1.5 m in Hudson Bay, and 2 m in Foxe Basin (Martini, 1986). In Hudson Bay, peak inputs from SIM are from June to mid-July, and during this time SIM usually provides more freshwater to surface waters than river runoff

(Prinsenberg, 1988; Granskog et al., 2011). The pulse of melt water also creates strong vertical stratification, which suppresses mixing of heat and nutrients into the surface waters (Prinsenberg, 1988; Else et al., 2008). Ice formation is also responsible for brine formation due to salt rejection (Saucier et al., 2004, Granskog et al., 2011).

### 3. Methods

We collected samples at 55 stations, including 16 rivers, across the Hudson Bay system from July 7$^{th}$ to 30$^{th}$, 2010, during leg 1a

of the 2010 ArcticNet Cruise aboard the CCGS Amundsen (Figure 1). Water samples for salinity, DIC, TA, and stable isotopes of both oxygen in seawater ($\delta^{18}$O) and carbon ($\delta^{13}$C$_{DIC}$) were collected at various depths using 12-L Niskin bottles mounted on a 24-bottle rosette fit with a SeaBird SBE911 Conductivity-Temperature-Depth (CTD) profiler. A small subset of samples collected in the Nelson River estuary, as well as river samples, were taken using either a single 3-L Niskin bottle hand-lowered to 1 m depth (for DIC, TA, and associated salinities), or using a bucket hand-lowered to less than 1 m depth (for $\delta^{18}$O, $\delta^{13}$C$_{DIC}$,

and associated salinities).

Samples for DIC and TA analysis were sampled directly from the Niskin bottles into borosilicate glass bottles: 250 mL or 500 mL bottles with ground-glass stoppers and elastic closures or 250 mL screw-cap bottles. All DIC and TA samples were poisoned with 100 µl of a saturated HgCl$_2$ solution to halt biological activity and were stored in the dark at either room temperature or at 4$^{o}$C until being processed ashore. The DIC and TA analyses were conducted at both Dalhousie University in

Halifax, Nova Scotia, and the Institute of Ocean Sciences (IOS) in Sidney, British Columbia. Both labs analyzed samples for DIC coulometrically followed by TA analysis from the same bottle using potentiometric titrations. At Dalhousie, both DIC and TA were analyzed using a VINDTA 3C (Versatile Instrument for the Determination of Titration Alkalinity, Marianda), whereas IOS analyzed DIC using a SOMMA (Single-Operator Multiparameter Metabolic Analyzer) and TA with a custom-built titration system. The analytical methods followed the recommendations of Dickson et al. (2007). Both labs used Certified Reference

Materials (CRM batches 101 and 81) supplied by A.G. Dickson (Scripps Institution of Oceanography) to consistently calibrate the instruments. Samples measured by the IOS lab have an analytical precision better than 0.9 µmol kg$^{-1}$ for DIC and 2 µmol kg$^{-1}$ for TA. An equivalent precision computation could not be done for samples analyzed at Dalhousie due to a lack of duplicate samples, but historically, precision of the Dalhousie VINDTA system has been consistently better than 3 µmol kg$^{-1}$ (e.g. Shadwick et al., 2011). We computed the aragonite saturation state ($\Omega_{Ar}$) from the DIC and TA data using the *CO2sys*

program of Lewis and Wallace (1998), with the equilibrium constants of Mehrbach et al. (1973) refit by Dickson and Millero (1987).



At station 740 in central Hudson Bay (Figure 1), an inter-calibration between the IOS and Dalhousie labs was conducted wherein duplicate samples were taken from all 12 Niskin bottles, with each lab producing independent depth profiles of both DIC and TA. Anomalously high deviations in both DIC and TA were measured at the bottom-most sample, and CTD profiles at this station reveal distinct changes in temperature and salinity in the bottom four meters of the water column, suggesting the presence

of a bottom nepheloid layer. Given that the bottom Niskin bottle was closed within this layer, the large deviations in both DIC and TA between duplicate samples may be due to large geochemical gradients or high concentrations of suspended material in the Niskin bottle. Not including these bottom samples, mean absolute differences in DIC and TA samples between the two labs were $2.4 \pm 0.9$ µmol kg$^{-1}$ and $12.6 \pm 6.8$ µmol kg$^{-1}$, respectively. Deviations in TA between the 11 duplicate samples (bottom sample not included) were not consistently positive or negative, thus an offset correction could not be applied to either dataset

prior to merging. Although this average TA difference represents significant variability between the two datasets, TA alterations of this magnitude do not alter the key results of this study and thus are not important in the context of this discussion. For example, a $\pm 12.6$ µmol kg$^{-1}$ change in surface water TA at all stations results in an average change in surface water $\Omega_{Ar}$ of $\pm 0.12$, with no visible change in spatial patterns throughout the Hudson Bay and with no stations moving from supersaturation to undersaturation, or vice versa. Also, patterns in $\Omega_{Ar}$ along the coastal transect in Hudson Bay, including the depth of the

saturation horizon, do not change significantly given a 12.6 µmol kg$^{-1}$ change in TA. Furthermore it has to be noted that the sampling strategy used in this study was not ideal for intercalibration purposes, since it took place in highly variable relatively shallow waters as compared to stable deep ocean waters.

Samples for $\delta^{13}C_{DIC}$ analysis were collected in 30 mL vacuum-sealed glass vials and spiked with 60-µL of saturated HgCl$_2$ solution to halt biological activity. Analysis of $\delta^{13}C_{DIC}$ was conducted at Yale University using continuous-flow isotope-radio-

monitoring mass spectrometry (CF-irmMS) on Thermo Finnigan MAT 253 gas mass spectrometers coupled to a Thermo Electron Gas Bench II via a Thermo Electron Conflo IV split interface. The analytical method has a reproducibility of better than $\pm 0.1$ ‰. Samples for the oxygen isotopic composition of water ($\delta^{18}O$) were collected in 20 mL borosilicate glass vials, sealed with Parafilm to minimize evaporation, and stored at 4$^o$C. Analysis was conducted at the G.G. Hatch Stable Isotope Laboratory (University of Ottawa) using a Finnigan MAT Delta plus XP + Gasbench. A 0.2 to 0.6 mL subsample was flushed with 2 % CO$_2$

in helium. For more information regarding this method, see Friedman et al. (1977). Stable carbon and oxygen isotope ratios are expressed in the usual delta ($\delta$)-notation as per mil (‰) deviation relative to the international VPDB (Vienna Pee Dee Belemnite) and V-SMOW (Vienna Standard Mean Ocean Water) standards, respectively. Salinity samples were analyzed on board the Amundsen using a Guildline 8004 B Autosal Laboratory salinometer calibrated with standard seawater from the International Association for the Physical Sciences of the Oceans (IAPSO).

The fractions of sea-ice melt water (f_SIM) and meteoric water (f_MW) are calculated using bottle salinities and $\delta^{18}O$ in a three end-member mixing model. The methods, equations, and end-member values (for seawater, sea-ice melt water, and meteoric water) used in the model are identical to those described by Granskog et al. (2011).





## 4. Results and Discussions

### 4.1 Surface Distributions

Surface distributions within Hudson Bay (Figure 2) reveal the dominant role that freshwater input plays in altering the carbonate system parameters of surface waters, particularly in the southern and eastern parts of the bay. Distributions of TA (Figure 2b),

DIC (Figure 2c), and $\Omega_{Ar}$ (Figure 2d) mimic that of salinity (Figure 2a), with maxima in the high salinity waters of the Hudson and Foxe Straits, and minima along the southern coast of Hudson Bay. One notable exception however, is near the Nelson River (Figure 1), where stations exhibit low salinities but relatively high DIC, TA and $\Omega_{Ar}$. The dilution of carbonate parameters by freshwater is more pronounced for sea-ice melt (DIC and TA 300-600 µmol kg$^{-1}$, e.g., Miller et al., 2011) compared to river outflow (river DIC and TA ~700-1800 µmol kg$^{-1}$, Azetsu-Scott et al., 2014, and this study), and thus, distributions of DIC, TA,

and $\Omega_{Ar}$ in the southern Hudson Bay likely reflect the variable impacts of river runoff and sea-ice melt on the carbonate system.

The Nelson has the highest freshwater discharge of any single river in the Hudson Bay system, and nearly half of the river runoff into Hudson Bay enters via James Bay. These hydrographical features are well illustrated by the distribution of meteoric water fractions (f_MW) throughout the region (Figure 2f). In contrast, the fractions of sea-ice melt (f_SIM) are highest along the southern coast of Hudson Bay (Figure 2e), near the comparatively small Severn and Winisk rivers (2-6 times smaller discharge

than the Nelson, Dery et al., 2005). This region of high f_SIM corresponds well to the observed sea-ice distributions shown in Figure 1. It is important to note that the high positive f_SIM observed near the Nelson River is due to the use of a single flow-averaged $\delta^{18}O$ end-member for meteoric water (-14 ‰, Granskog et al., 2011) near a river with a considerably higher $\delta^{18}O$ (-10.8 ‰, Table 1). Also note that despite considerable freshwater input near James Bay, aragonite remains supersaturated ($\Omega_{Ar}$ > 1) at all stations in surface waters (Figure 2d).

The surface distributions shown in Figure 2 also align with the general circulation patterns of the Hudson Bay system. For example, differences can be seen between stations in the northern part of Hudson Strait, where Labrador Sea waters with characteristically high salinity, DIC and TA flow into the Hudson Bay system, and stations at the southern side of the strait, where Hudson Bay outflow with lower salinity and TA, as well as higher f_SIM and f_MW, travels east towards the Labrador Sea. The high salinity water in northwestern Hudson Bay represents water recently introduced from the Foxe and Hudson Straits,

with salinities generally decreasing in a counterclockwise fashion due to freshwater inputs. At the mouth of James Bay, the salinity and carbonate parameters are higher on the western edge, where waters enter the bay, compared to the eastern edge, where waters altered by large James Bay rivers exit back into Hudson Bay. At the eastern edge of the mouth of James Bay, we recorded the minimum salinity (S = 21.71), TA (1705 µmol kg$^{-1}$), and DIC (1624 µmol kg$^{-1}$), as well as the second lowest $\Omega_{Ar}$ (1.04), among the "seawater" samples from our entire study.

The surface distributions of carbonate parameters throughout the Hudson Bay system are similar to those reported by Azetsu-Scott et al. (2014) for September, 2005. Concentrations within Hudson Strait and Foxe Strait are comparable, and the north-south gradient within Hudson Strait is apparent in both years. Within Hudson Bay, concentrations are distinctively lower than in waters outside the bay, with values decreasing further from the northwest toward the southeast. Aragonite saturation ($\Omega_{Ar}$) in surface waters does not show any significant differences between 2005 and 2010, with surface waters approaching saturation at the

mouth of James Bay in both years. However, subtle differences do exist between these datasets. Stations in southern Hudson Bay have lower salinity, DIC, TA and $\Omega_{Ar}$ in 2010, whereas stations located further downstream, near the eastern edge of the bay,



have lower values in 2005. Rather than reflecting basin-wide changes over the 5-year period, these differences can be ascribed to the seasonal timing of sample collection and general circulation patterns. That is, stations along the southern coast are impacted by substantial SIM in July as the remaining ice is melting, and the cyclonic circulation is capable of transporting such low-salinity water along the eastern edge of Hudson Bay over a two-month period (i.e., from July to September) (cf. Granskog et al.,
5    2009).

### 4.2 Hudson Bay Source Waters

The main conduits for Pacific and Atlantic waters to the Hudson Bay System (HBS) are Fury and Hecla Strait and Hudson Strait, respectively. Yet it remains unclear whether Hudson Bay is composed primarily of Pacific or Atlantic waters. Jones et al. (2003) surmised that Pacific waters were more prevalent in Hudson Bay but were unable to provide conclusive evidence, due to
limitations of the quantity and quality of their data. In regions like Hudson Bay, with multiple seawater and freshwater end-members, mass-balance calculations for unravelling freshwater components are particularly complicated, and in contrast to systems such as the coastal Beaufort Sea, where three rather distinct end-members (the Mackenzie River, sea-ice melt, and a primarily Pacific polar mixed layer) can be defined by two tracers (Macdonald et al., 1995), such an analysis in Hudson Bay requires additional tracers.

When examining the relationship between TA and salinity throughout the HBS (Figure 3a), waters below 100 m depth in Hudson Bay are indistinguishable from those in Foxe Strait, and both are distinctly different from deep waters in Hudson Strait. Furthermore, data from Hudson Bay and Foxe Strait are well aligned with the Pacific water end-member, while data from Hudson Strait falls along a mixing line with the Atlantic water end-member characterized by Shadwick et al. (2011) (Figure 3a). This indicates that deep water from Hudson Strait does not enter Foxe Strait and that the vast majority of water that flows over
the 185 m sill into Hudson Bay is likely of Pacific origin. In addition, some samples in the Hudson Bay/Foxe Strait dataset have higher salinity and TA than the Pacific water end-member, yet still fall along the extension of the general mixing line (Figure 3a) suggesting an additional source of both TA and salinity. One plausible source of excess TA and salinity in deep waters is from brine rejection. It is known that deep-water formation occurs in Foxe Basin polynyas, and this water can flow into Foxe Strait and potentially further into Hudson Bay (Defossez et al., 2010). However, there is evidence in Hudson Bay that deep waters can
also form locally due to brine rejection (Granskog et al., 2011). The same general patterns are shown in a plot of deep water DIC against salinity (Figure 3b), except that distinctly higher DIC is observed in Hudson Bay compared to Foxe Strait due to the build-up of respiratory DIC in Hudson Bay, which is discussed in greater detail below.

### 4.3 Freshwater inputs

Prior studies have revealed the importance of watershed geology in governing the highly variable compositions of Hudson Bay
rivers (Mundy et al., 2010; Granskog et al., 2011; Azetsu-Scott et al., 2014, see also Thomas and Schneider, 1999), yet relatively little is known regarding the impact of varying river inputs on the inorganic carbon system of Hudson Bay. Variations in TA with salinity in the top 60 m of the water column of Hudson Bay (Figure 4) reveal two distinct freshwater end-members: a lower TA end-member of ~700 $\mu$mol kg$^{-1}$, corresponding to the vast majority of stations in the bay, and a high TA end-member of ~1900 $\mu$mol kg$^{-1}$ for samples collected within the Nelson River estuary (NRE, Figures 1 and 5). Figure 4 also highlights the
various impacts of sea-ice melt, as samples with high f_SIM (purple circles) show distinct positive deviations from the $\delta^{18}$O vs. salinity mixing line (Figure 4, inset), as well as negative deviations from the TA vs. salinity mixing line, due to the enriched $\delta^{18}$O signature ($\delta^{18}O_{SIM}$ = 0 ‰) and low TA concentration characteristic of sea-ice melt.





Considering that samples with potential inputs of brine (i.e., deeper waters) and discernable sea-ice melt input (i.e., stations with f_SIM ≥ 0.05) are not included in the best-fit regressions, the lower end-member (that is, 689 µmol kg$^{-1}$) likely represents the average riverine TA for the entire Hudson Bay. This end-member aligns with the prior estimate for "rivers" of 754 µmol kg$^{-1}$, also derived from regression lines of samples collected within the bay and not from the rivers, themselves, along the eastern

shore of Hudson Bay (Azetsu-Scott et al., 2014). Stations in southwestern Hudson Bay, nearest to the Churchill, Nelson, and Hayes rivers (Figure 4, red squares, '705' stations and further North, station 706, see Figures 1 and 5), have a similar salinity-TA relationship to that observed throughout Hudson Bay, but are vertically offset to a slightly elevated TA. This TA offset is not related to the merging of the two TA datasets, as the stations labelled as "SW Hudson Bay" (see Figure 4) comprise a small subset of the TA data analyzed at IOS. Also, the magnitude of this vertical TA offset (~80 µmol kg$^{-1}$) is much larger than the

average deviation observed between the datasets. Instead, this offset likely indicates a benthic carbonate input from the sediments. For salinity vs. DIC, the same offset is visible but is less pronounced, which also points towards benthic carbonate addition. Azetsu-Scott et al. (2014) attributed their observations of high TA in the deep waters of southwestern Hudson Bay to dissolution the carbonate-rich bedrock (defined as the Hudson Bay Lowland) that underlays this area of the bay.

We attribute the high TA end-member (1870 µmol kg$^{-1}$) to the Nelson River. A similarly high TA end-member has been reported

for the Mackenzie River (1540 µmol kg$^{-1}$, Cooper et al., 2008) and time-series measurements collected from the Churchill River (Figure 1) during the summer of 2007 reveal an average TA of 1394 ± 80 µmol kg$^{-1}$ (Stainton, 2009). According to Mundy et al. (2010), the Churchill, Nelson, and Hayes rivers (Figure 1) have dissolved organic carbon (DOC) concentrations that are 2-6 times greater than other Hudson Bay rivers, and these authors attributed variability in riverine DOC to the vastly differing watershed geologies throughout the Hudson Bay region. The weathering of organic rich soils throughout the extensive Nelson

River drainage basin would contribute to higher riverine DIC and TA compared to rivers further North with drainage basins in the relatively organic-poor Arctic tundra. Furthermore, weathering processes become more active in subarctic, or even temperate regions of the Hudson Bay drainage area, yielding higher TA inputs from those regions than from polar regions. This regional dependence on riverine composition is also observed in δ$^{18}$O signatures, with rivers along the southern coast having distinctly less depleted δ$^{18}$O signatures compared to rivers further north (Table 1, see also Granskog et al., 2011). Nevertheless, the TA

end-member calculated here is substantially higher than the prior estimate for the Nelson River (1022 µmol kg$^{-1}$, Azetsu-Scott et al., 2014), and we think this discrepancy is due to sampling depth within this highly-stratified system, as well as to the fact that our samples extended further up the estuary, into fresher waters.

Data obtained via rosette sampling in southwestern Hudson Bay (Figure 4, red squares) yield a riverine TA end-member of 710 µmol kg$^{-1}$, and the uncertainty on this value places it in the same range as the end-member calculated for the rest of Hudson

Bay. According to the CTD data, samples from southwestern Hudson Bay were collected between 1.4 m and 10.3 m depth, with the former being considered 'surface' samples. In contrast, samples collected from the surface in the inner NRE using the small barge yield an end-member of 1870 µmol kg$^{-1}$. Given the close proximity of these inner NRE stations to others in southwestern Hudson Bay, the discrepancy in end-members indicates that surface samples collected via the rosette capture a different water mass compared to samples captured via the small barge. This is illustrated further using a cross-section of salinity in the NRE

(Figure 5).



Salinity across the NRE, as measured in-situ by the CTD attached to the rosette system, and as measured from discrete surface bottle samples collected either from the barge or using a bucket dropped from the bow, is shown in Figure 5. At stations 705a, b, and c, salinity samples collected from surface waters via bucket have considerably lower salinities than any waters measured by the CTD. Furthermore, according to the CTD pressure data, surface Niskin bottles from stations 705a, b, and c were closed at

1.6, 1.6 and 1.4 m respectively (Figure 5, black X's), but the salinities measured in these Niskin bottles correspond to the CTD salinities observed at depths of 2.5, 3.6 and 7.1 m respectively (Figure 5, vertical black arrows), indicating that the Niskin bottles contained some deeper waters entrained into the surface by the upward-moving rosette. This highlights the importance of taking bottle salinity samples from Niskins in parallel with parameters such as DIC and TA, particularly in highly stratified waters. Here, all DIC and TA samples collected in southwestern Hudson Bay are accompanied by a high-precision salinity sample taken

from the same Niskin bottle.

At stations B6, B8, and B12 (Figure 5), salinities measured using the 3-L Niskin bottles (deployed to 1 m depth from the small barge) are very low and show an increasing salinity with distance away from the Nelson River mouth that is consistent with the bucket samples collected at stations 705a, b and c. Clearly, the samples collected from the small boats (barge and zodiac) captured low-salinity waters very near the surface, while the 'surface' rosette bottles captured water from a deeper layer with

considerably higher salinities. Assuming the low salinity layer represents the Nelson River plume, this water is only accurately captured using the samples taken via small boats, which explains why samples taken via barge in the inner NRE have a markedly different TA vs. salinity relationship from samples taken via rosette throughout the remainder of Hudson Bay (Figure 4). Additionally, at station 705c (Figure 5), the bucket sample near the surface does not appear to have captured the low-salinity river plume, demonstrating the extent to which river inputs are constrained to the coastline during cyclonic transport around the

bay (Figure 1). Large CTD salinity gradients in the upper 2 m were observed at other stations along the southern coast, suggesting that data from the surface Niskin bottle at these stations may also not fully reflect the true "surface water" properties.

### 4.4 Coastal Transect:

Hudson Bay waters undergo significant compositional changes during their cyclonic transit around the basin (Granskog et al., 2009; Mundy et al., 2010). Here, we focus specifically on changes in the carbonate system along a coastal transect that follows

the general counter-clockwise circulation pattern from Northwest Hudson Bay to the Northeast (Figure 6, transect A-B). For stations within this transect, TA, DIC and $\delta^{13}C_{DIC}$ are plotted against in-situ density anomaly ($\sigma_t$, observed density – 1000 kg m$^{-3}$) to illustrate deviations at constant density levels (Figures 6a-c). For TA (Figure 6a), there is little variability between stations along the transect and the vast majority of the data fall along the river mixing line. There are, however, two notable exceptions: the first being high-TA deviations at stations within the NRE that we attribute to benthic carbonate addition, and the second

being low-TA deviations in samples with high sea-ice melt fractions ($f\_SIM \geq 0.05$), all of which are measured at or near the surface along the southern coast.

For along-transect DIC (Figure 6b), the impacts of both freshwater input and biological activity are apparent. Similar to TA, positive deviations from a general river mixing line are observed in the NRE, as are negative deviations along the southern coast, but a distinct increase in DIC is also visible along the transect due to a build-up of respiratory DIC in subsurface waters. The

maximum offset between waters along the west coast (blue-purple points) and the east coast (red-yellow points) is between the 24 and 26 kg m$^{-3}$ $\sigma_t$ horizons, indicating that organic matter respiration rates are highest at these density levels. The structure of



these density horizons along the coastal transect are shown in Figure 6d. The lack of any visible offset in bottom water TA suggests that any products of anaerobic respiration that may be produced in sediments are not transported into the water column.

Unlike TA and DIC, freshwater input appears to play a relatively minor role in governing variability in the stable carbon isotope signature of DIC ($\delta^{13}C_{DIC}$) of Hudson Bay waters (Figure 6c). This may be due to the fact that Hudson Bay rivers are very

isotopically enriched (average $\delta^{13}C_{DIC}$ = -3.2 ‰, Table 1) compared to other coastal regions (North Sea $\delta^{13}C_{DIC}$ = -12 to -16 ‰; Burt et al., 2016), and thus are much closer to typical seawater values (~0-2 ‰). Biogenic soils are highly depleted in $\delta^{13}C_{DIC}$ (~ -25 ‰), while carbonate bedrock has a $\delta^{13}C_{DIC}$ near 0 ‰ (Spiker, 1980). Given that the majority of Hudson Bay rivers have rather low DOC (Mundy et al., 2010) and that much of the bedrock surrounding the bay is carbonate-rich (Ross et al., 2011), it is unsurprising that Hudson Bay rivers are isotopically enriched compared to other regions. However, $\delta^{13}C_{DIC}$ was not sampled

along the southern coast, thus the Churchill, Nelson, and Hayes Rivers, which are likely more depleted in $\delta^{13}C_{DIC}$, are not included in Table 1 and are not well represented in Figures 6c and d.

The $\delta^{13}C_{DIC}$ distributions in Figure 6c highlight the impact of biological activity along the coastal transect. Waters become isotopically lighter during their transit around the basin, because respiration of organic matter releases isotopically light DIC into the water column. In accordance with Figure 6b, relative changes between waters along the western shore (blue-purple points)

and eastern shore (red-yellow points) are greatest in intermediate waters. In the North Sea, Burt et al. (2016) showed that a respiratory DIC increase of 1 μmol kg$^{-1}$ could be roughly equated to a 0.012 ‰ decrease in $\delta^{13}C_{DIC}$. Taking this ratio, and assuming that waters are transported along lines of constant density, changes in $\delta^{13}C_{DIC}$ at intermediate depths (between ~25-50 m, at $\sigma_t$ = 25 kg m$^{-3}$, see Figures 6c and d) between western and eastern waters equate to an addition of approximately 110 μmol kg$^{-1}$ DIC. Observed increases in DIC at the same density level (Figure 6b) are approximately 90 μmol kg$^{-1}$, which is

comparable given the rough nature of this calculation. In deeper waters (between ~40-100 m, at $\sigma_t$ = 26 kg m$^{-3}$, see Figure 6d), both observed DIC increases shown in Figure 6b, and increases based on differences in $\delta^{13}C_{DIC}$ (shown in Figure 6c) equate to ~70 μmol kg$^{-1}$.

The effects of biological activity, sea-ice melt, and variable riverine input on the carbonate system of Hudson Bay are well summarized in the coastal transect of aragonite saturation state ($\Omega_{Ar}$) shown in Figure 6e. In surface waters, $\Omega_{Ar}$ is high in the

Northwest but decreases along the southern coast due to the dilution of both TA and DIC by freshwater input. In the NRE, $\Omega_{Ar}$ remains high despite low salinities (see Figure 2a) because carbonate addition from the seafloor adds TA and DIC in a ratio of 2:1, thereby counteracting the dilution effect of river input (Figure 4). Surface waters near James Bay show minima in $\Omega_{Ar}$, likely due to the strong sea-ice melt signal in these waters, as well as large inflows of low-alkalinity river waters (see Figure 2). Along the eastern shore, surface water $\Omega_{Ar}$ is comparable to (i.e., only slightly lower than) values along the west coast. This is likely due

to the fact that the sea-ice melt waters, which dilute TA and DIC to a much greater degree than river input, have yet to be transported to these stations. In September 2005, surface water $\Omega_{Ar}$ along the southern coast was higher than observed here, while further downstream along the eastern coast, $\Omega_{Ar}$ was lower than observed here (Azetsu-Scott et al., 2014). This may simply reflect the transport of low-$\Omega_{Ar}$ sea-ice melt waters from the southern coast in July to the eastern shore by September. In accordance with this, Granskog et al. (2009) noted that in October 2005, virtually no sea-ice melt was observed along the

southern shore, further suggesting that these waters had been transported northeast by the coastal current.



Focusing on subsurface waters along the transect, the saturation horizon shoals significantly from 60-70 m along the west coast, to 30-40 m along the east coast. Furthermore, deep waters along the west coast are only slightly undersaturated, while deep waters along the east coast are heavily undersaturated. These patterns likely reflect the build-up of respiratory DIC in deep waters during transit around Hudson Bay, as aerobic respiration releases DIC while consuming TA, thus causing a relatively

strong decrease in $\Omega_{Ar}$. Deep waters along the eastern shore are likely older than waters along the western shore, allowing the products of aerobic respiration to build-up over time. Exchange with open ocean waters is limited by the shallow sills at the mouth of Hudson Bay, and very low saturation states are observed at deeper stations in the central Bay, as well as in the coastal transect (Figure 6e).

Subsurface $\Omega_{Ar}$, both along the coastal transect (Figure 6e) and in a cross-basin section (not shown), is very similar to the 2005

distributions shown by Azetsu-Scott et al. (2014), suggesting that the acidification state of Hudson Bay has not changed discernibly between 2005 and 2010. However, with only two such datasets taken at slightly different times of the year, more information is required to make conclusive statements regarding acidification rates in this region.

**5. Conclusions:**

This work contributes insight into important carbon cycling processes in a region of the Arctic where biogeochemical data are sparse. The Hudson Bay is home to a vast number of ecologically important species, thus an improved understanding of the key processes affecting the carbonate system, and thereby the ecosystem, in the bay is crucial. Surface distributions reveal the importance of variable freshwater inputs from runoff and sea-ice melt in altering the carbonate system, especially in the southern part of Hudson Bay. Rivers draining into Hudson Bay have highly variable DIC and TA, a highly-enriched $\delta^{13}C_{DIC}$ signature

compared to other regions, and $\delta^{18}O$ signatures that show a strong regional dependence. This study also shows that the deep Hudson Bay is a) primarily filled with waters of Pacific origin from the Canadian Arctic Archipelago and b) thus has limited interaction with Atlantic waters in the deep Hudson Strait, which supports preliminary findings by Jones et al. (2003). Within Hudson Bay, the salinity vs. TA relationship indicates mixing between the Pacific waters and river waters with relatively low TA, compared to rivers at lower latitudes. Negative deviations from the mixing line are due to dilution from sea-ice melt,

whereas positive deviations in the Nelson River estuary point to benthic carbonate addition. Direct sampling of river DIC and TA throughout the bay, as well as detailed surface water sampling (i.e., from multiple platforms) in highly stratified estuarine waters, are needed to better understand exactly how river inputs alter the carbonate system parameters of Hudson Bay. This is especially important given that river input into Hudson Bay is likely changing significantly due to climate change and, perhaps more importantly, continued hydroelectric development in the region. Finally, $\Omega_{Ar}$ in deep waters show no discernable change from

2005 levels, and slight changes in surface water $\Omega_{Ar}$ can be attributed to seasonal variability.

Author Contributions: W.J. Burt – Primary data analysis, primary writer and editor of manuscript. H. Thomas – assistance of data analysis, writing and editing. L. Miller – collection and analysis of IOS data, assistance of data analysis, writing and editing. M. Granskog, T. Papakyriakou, and L. Pengelly – editorial assistance.





Acknowledgements - *Thank you to the ArcticNet community for their support and helpful feedback. Also, thanks to Conrad Koziol for collecting the DIC and TA data during leg 1a and to Marty Davelaar for sample analyses, as well as the captain, crew and the ArcticNet scientists abroad the CCGS Amundsen. M. A. Granskog was supported by the Centre for Ice, Climate and Ecosystems at the Norwegian Polar Institute and the Fram Centre Ocean Acidification Flagship research program funded by the Ministry of Climate and Environment of Norway. This research was supported through grants from NSERC and ArcticNet.*



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



**Tables**

**Table 1. Stable Isotope data from Hudson Bay rivers.**

| River | Date | Salinity | $\delta^{18}O$ | $\delta^{13}C_{DIC}$ |
|---|---|---|---|---|
| **Northeast** | | | | |
| Polemund | 10-Jul-10 | n/a | -14.53 | -1.82 |
| Povungnituk | 10-Jul-10 | n/a | -15.69 | -4.13 |
| Kogaluc | 10-Jul-10 | n/a | -14.58 | -2.56 |
| Innuksuac | 11-Jul-10 | n/a | -13.85 | n/a |
| **Southeast** | | | | |
| Nastapoca | 12-Jul-10 | 0.02 | -14.27 | -2.86 |
| Little Whale | 12-Jul-10 | n/a | -14.48 | -3.69 |
| Great Whale | 13-Jul-10 | 0.33 | -14.17 | n/a |
| **Northwest** | | | | |
| Wilson | 18-Jul-10 | n/a | -14.93 | -2.95 |
| Ferguson | 18-Jul-10 | n/a | -16.22 | -2.93 |
| Tha-anne | 19-Jul-10 | n/a | -16.82 | -3.56 |
| Thlewiaza | 19-Jul-10 | n/a | -16.08 | -4.18 |
| **Southwest** | | | | |
| Churchill | 20-Jul-10 | n/a | -12.71 | n/a |
| Severn | 28-Jul-10 | n/a | -11.01 | n/a |
| Winisk | 28-Jul-10 | n/a | -10.76 | n/a |
| Nelson | 27-Jul-10 | n/a | -10.81 | n/a |
| Hayes | 27-Jul-10* | n/a | -11.45 | n/a |

*exact timestamp not recorded

n/a = data not available



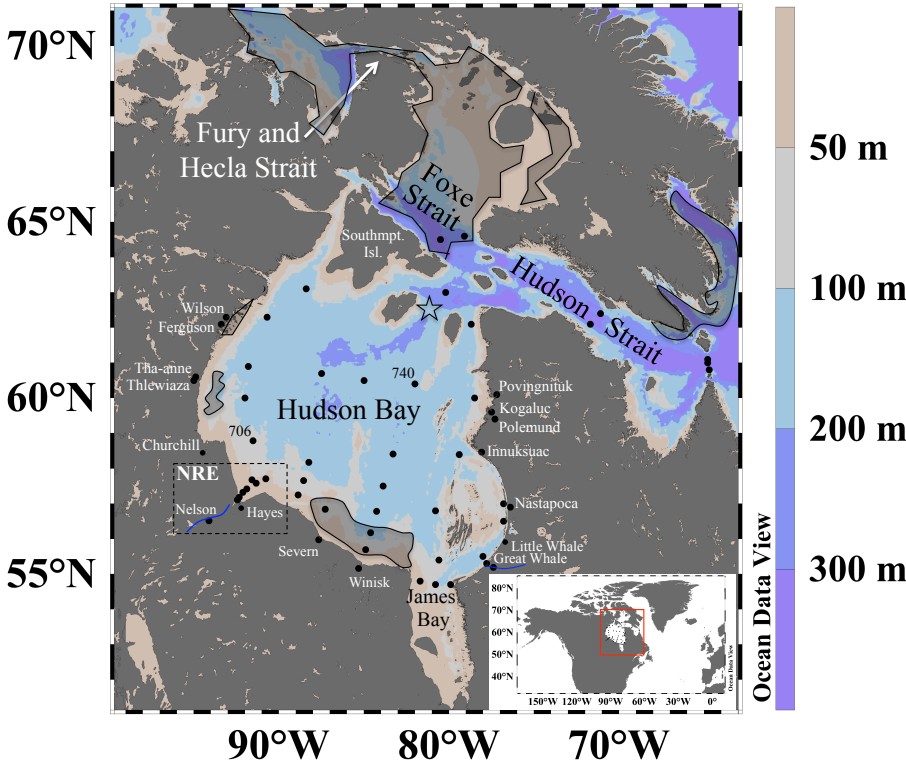

**Figure 1: Bathymetric map of the Hudson Bay System showing stations sampled during 2010 ArcticNet mission (black dots). Shaded areas denote ice-covered regions as of July 12[th], 2010 determined using weekly regional ice extent maps (Environment Canada, Canadian Ice Service, http://www.ec.gc.ca/glaces-ice/). The location of the 185 m sill in the central channel connecting Hudson Bay with Foxe Strait and Hudson Strait is marked with a black star.**





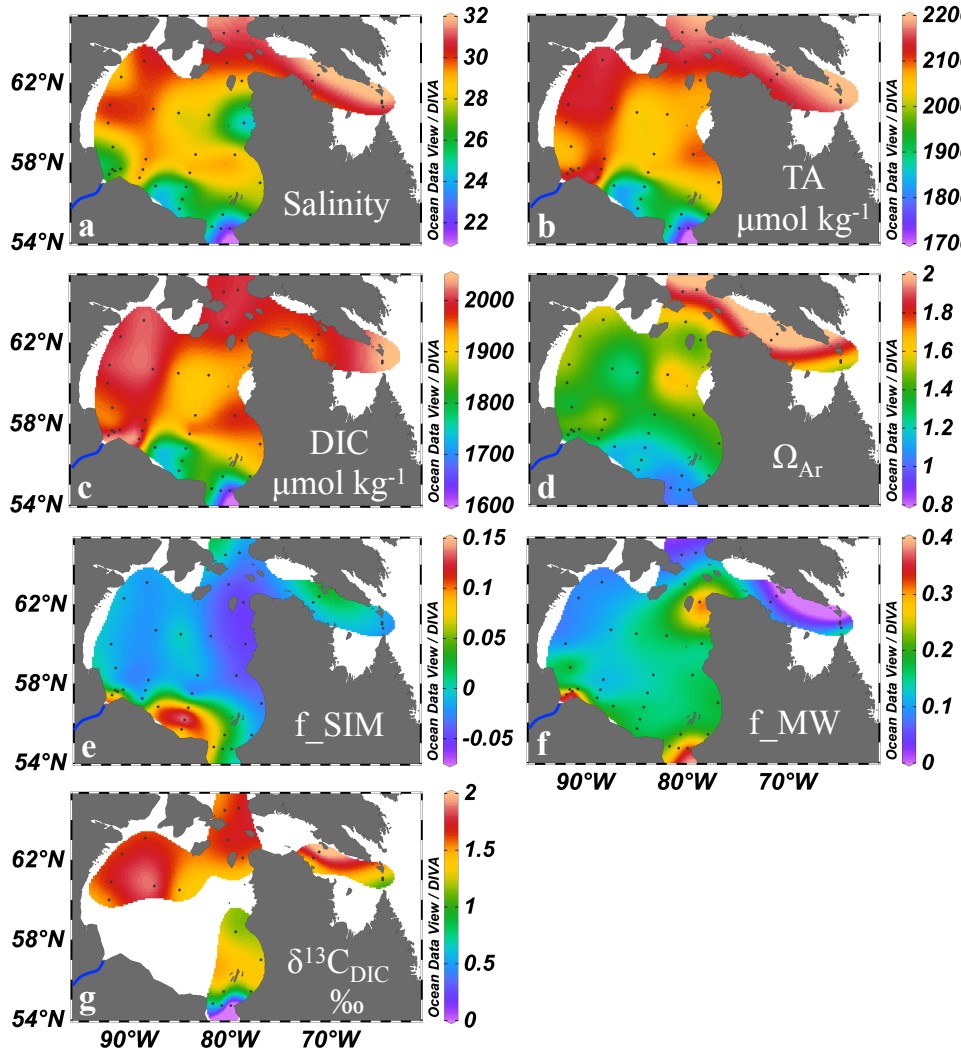

**Figure 2: Surface distributions of salinity (a), TA (b), DIC (c), aragonite saturation state ($\Omega_{Ar}$) (d), sea-ice melt fraction (f_SIM) (e), meteoric water fraction (f_MW) (f), and $\delta^{13}C_{DIC}$ (g). Nelson River is shown (blue line). Note that $\Omega_{Ar}$ (panel d) is above one at all stations, thus no contours are shown.**



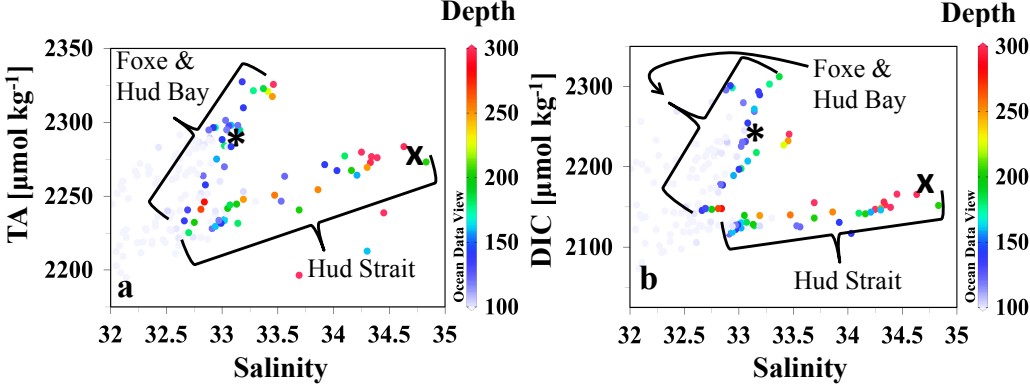

**Figure 3: Deep water TA (a) and DIC (b) against salinity throughout the Hudson Bay System. The \* represents the end-member properties of Pacific-origin Upper Halocline Water (S = 33.1, TA = 2283 µmol kg⁻¹, DIC = 2236 µmol kg⁻¹) and the black 'x' represents Atlantic Water (S = 34.8, TA = 2301 µmol kg⁻¹, DIC = 2154 µmol kg⁻¹), as defined by Shadwick et al. (2011). Hud Bay = Hudson Bay samples, Foxe = Foxe Strait samples, Hud Strait = Hudson Strait samples.**




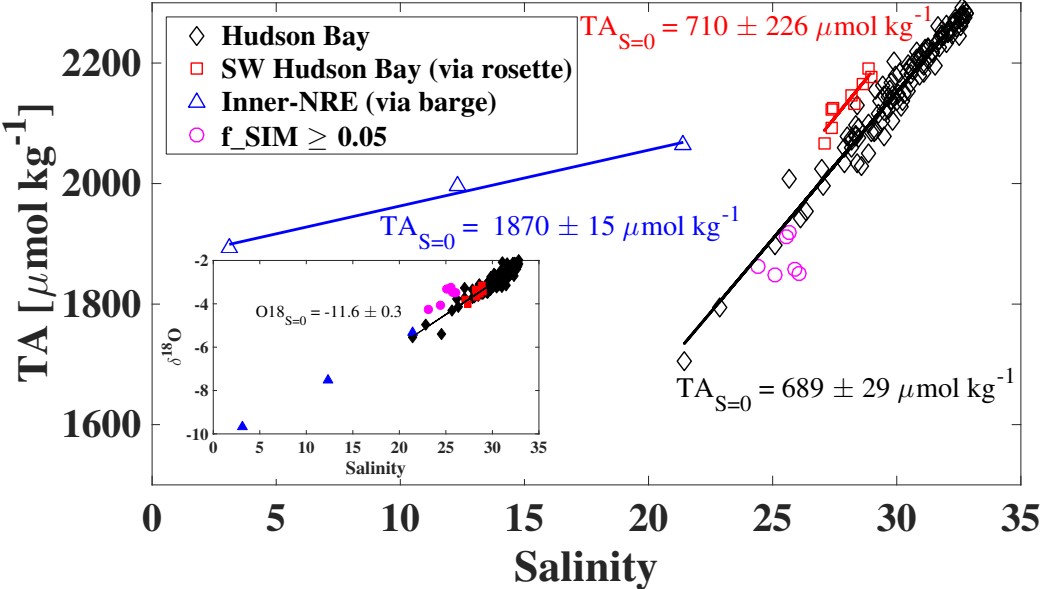

**Figure 4:** Hudson Bay TA vs. Salinity (S) in the upper 60 m of the Hudson Bay water column (which should contain seasonal freshwater inputs, see Granskog et al., 2011). Hudson Bay stations (black diamonds, regression line TA = 48.8*S + 689.2) do not include the station located northeast of the sill in the central channel (see Figure 1). Stations in southwestern Hudson Bay (red squares, regression line TA = 50.9*S + 225.8) refer to '705' stations (see Figure 5), and station 706 slightly further north (see Figure 1), while Inner-NRE stations (blue triangles, regression line TA = 9.3*S + 1870.0) refer to 'B' stations shown in Figure 5. Riverine end-members (TA at zero-salinity, or $TA_{S=0}$) and their corresponding uncertainties are calculated using linear least-squares regression and the error associated with the linear fit, respectively. Inset: Hudson Bay $\delta^{18}O$ vs. Salinity with same sample legend as the main figure (symbols are filled for clarity). The regression line ($\delta^{18}O = 0.3*S - 11.6$) is placed through Hudson Bay samples. It is important to note that for some surface water samples, $\delta^{18}O$ and its corresponding salinity were collected independently from TA and its corresponding salinity, therefore high f_SIM (calculated using $\delta^{18}O$) may not be present in TA samples. This may explain why some samples with high f_SIM show little to no negative deviation in TA.



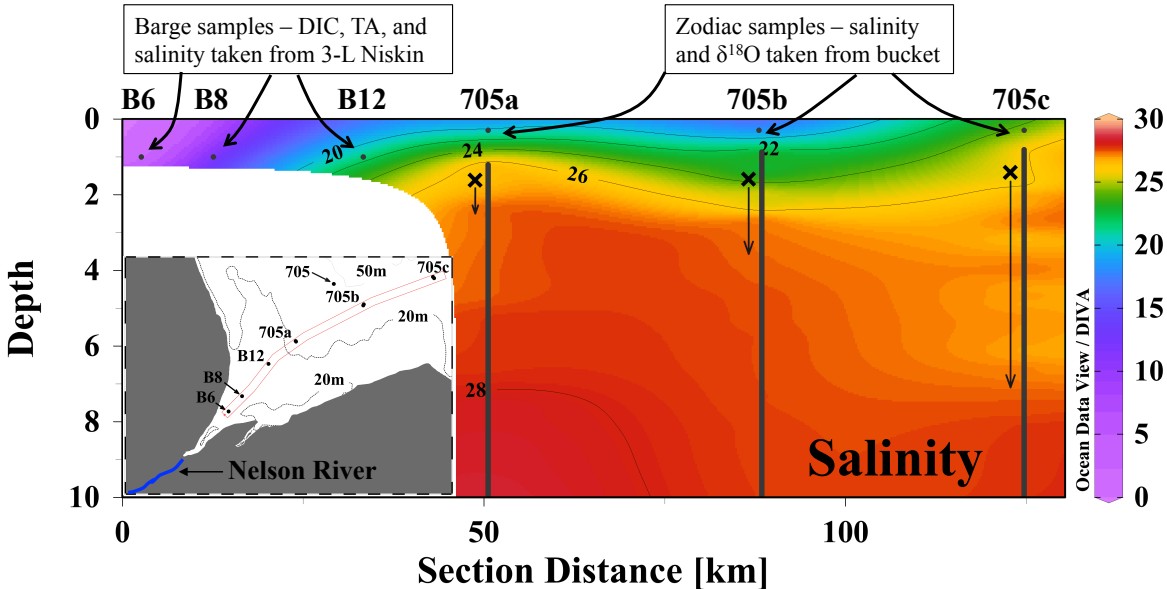

**Figure 5: Salinity cross-section (top 10 m) in the Nelson River estuary (NRE). Location of transect is shown by the inset map in the lower right (see Figure 1 for location of NRE within Hudson Bay). Vertical black lines represent CTD profiles, while black dots (depths ≤ 1 m) represent samples taken independently using the single 3-L Niskin (~1 m depth, stations B6, B8, B12) and samples taken using the bucket (~ 0.3 m depth at stations 705 a,b,c). Black X's show the depth where surface Niskin bottles were closed, as recorded by the CTD's pressure sensor, while the black vertical arrows extend to the depth where salinities in the Niskin bottles match the salinity recorded by the CTD. At all three stations (705 a,b,c), Niskin bottle salinities are markedly higher than the CTD salinities at the closure depth, suggesting that higher-salinity waters were entrained into surface waters by the rosette system.**





**Figure 6:** Variations in TA (a), DIC (b) and $\delta^{13}C_{DIC}$ (c) with in-situ density anomaly ($\sigma_t$, observed density – 1000 kg m$^{-3}$) along the Northwest-Northeast (A-B) coastal transect (inset map of transect is shown). Dashed ellipses surround data within the NRE, while solid ellipses surround data with f_SIM ≥ 0.05 (located along southern coastline). Vertical arrows illustrate differences in DIC and $\delta^{13}C_{DIC}$ between waters in western Hudson Bay (blue/purple points) and eastern Hudson Bay (red/yellow points). d: Cross-section of $\delta^{13}C_{DIC}$ along the coastal transect with isopycnals of in-situ density anomaly ($\sigma_t$, kg m$^{-3}$) overlain. e: Cross-section of $\Omega_{Ar}$ with aragonite saturation horizon ($\Omega_{Ar} = 1$) overlain.