# Peer review of "Inorganic Carbon Cycling and Biogeochemical Processes in an Arctic Inland Sea (Hudson Bay)"

_Biogeosciences, 2016_

## Referee Comment (RC1) · Anonymous Referee #1 · 7 Jun 2016

This work presents a nice data set from a region where few data has been collected, especially when focusing on the inorganic carbon cycle and how this is impacted by input from land. It is largely well written and the figures illustrate the results. However there are some minor issues that the authors should consider before publication.

One is an aspect that often is sees in the literature. The sentence that start in line 25 of page 1 reads that the Arctic Ocean is particularly vulnerable to OA because of . . . low buffer capacity in cold waters. . . The low buffer capacity means that DIC is not changing much for a given change in pCO2, which also means that the concentration of CO32- does not change as much as it will in warmer water of less equilibrium DIC concentration. Thus the saturation state of calcium carbonate changes less in cold water compared to warm water for a given increase in pCO2. So even if the saturation state is lower to begin with does this mean that these waters are more vulnerable? It

is best to be more specific when making this kind of statement.

A principle issue is that one analyse seawater but determine (or measure) TA, DIC or any other constituent. When analysing TA it means that one determines what TA consists of. One can claim that this is minor thing and that the word "analyse" often is used in this way, but that is not an excuse for not being correct.

Page 6, line 22. It is suggested that excess TA and salinity is added by brine, and on line 36 that sea ice melt result in a negative deviation of TA. This is highly likely considering the data and other observations from the Fox basin, but can one thus conclude that this result show that no ikaite (calcium carbonate) is formed and trapped in the sea ice. I would urge the authors to expand on this topic.

On page 7 the term "benthic carbonate input" is used. I assume that what is meant that metal carbonate dissolves at the sediment surface and that the carbonate ion is added to the bottom water. If so add "ion" after carbonate.

Next to last sentence of the first paragraph on page 8. It is extremely important to take samples for salinity determination from the same flask as where the chemical samples are collected in stratified waters. This cannot be stressed enough and this contribution again well illustrates this.

The section starting on line 12 at page 9. It would be interesting if the authors discuss how the observed C-13 signal is impacted by either decay of marine produced organic matter or terrestrial produced organic matter. Can the data be used to distinguish these sources?

Next paragraph. Here I have difficulties to understand how bottom water with high TA and DIC can be mixed up into the surface to give the high omega values without also mixing up salinity. Is it not more likely that the dissolution of metal carbonates occurs in the river drainage basin and is transported out to the estuary? Is there information on the mineral compositions of the different drainage basins that can add

to the observations?

---

## Referee Comment (RC2) · L. W. Cooper (Referee) · 10 Jun 2016

L. W. Cooper (Referee)

cooper@umces.edu

This is a short descriptive paper describing the results of sampling in 2010 for total alkalinity and inorganic carbon, including the isotopic composition of DIC in Hudson Bay, with the use of oxygen isotopes as tool to evaluate runoff and melted sea ice contributions. It follows related studies by Azetsu-Scott et al. 2014 and Granskog et al. 2011, among others in the same region, so it is of value both as a point in a time-series for possible changes in the inorganic carbon system and to fill in the geographical gap for arctic data coverage within Hudson Bay. Because of the limited one time nature of the sampling, it is not clear whether changes can be readily determined, but the scope of the paper seems appropriate for the descriptive nature of the data set presented. I recommend a few improvements and pose some questions below for the authors to consider.

Figure 1. The map could be improved. It is plotted in a Mercator projection, which is not a polar projection, so distances are distorted, and it is not clear where the latitude and longitude markers are. The shaded areas denoting sea ice are also rather awkwardly illustrated and hard to see, such as the northern straits connecting to Hudson Bay. I think this could be easily re-done in a standard GIS program in a more appropriate polar map projection with better labeling. The map figure in the Granskog et al. 2011 paper that is referenced in the text, while still plotted in Mercator projection, is a much more attractive example of how to display the geographical information that is important to the data presentation.

Figure 2, and 6c. Why isn't the $\delta$13C-DIC data coverage more complete? Again, as with Figure 1, the quality of these figures is only fair, and it could be improved by exporting out of Ocean Data View into a more robust graphics program, with appropriate labeling undertaken such as of the Nelson River (simply outlined in blue now). Also one of the software license requirements of Ocean Data View is to cite the software developers. "ODV may be used free of charge for non-commercial, non-military research and teaching purposes. If used in scientific papers, reports or posters, ODV must be referenced as follows: Schlitzer, R., Ocean Data View, odv.awi.de, 2015." Please add to the references or acknowledgements.

Figure 5 caption. The discrepancy with rosette bottle salinities can be caused by inadequate flushing time before bottles are closed at depth. Is the flushing time used by the CTD operator known?

Table 1. The one-time sampling of these rivers is a limitation for this type of study that depends upon integrated and flux weighted data. The table could be expanded and include references to any other available data for these rivers.

Page 4, line 25. Friedman and O'Neill (not Friedman et al. 1977) is really not a very appropriate reference for this method of determining oxygen isotopes. It describes the isotopic fractionation factors associated with equilibration of water with carbon dioxide,

but it is not really a description of the method. Epstein and Mayeda, 1953 in Geochemica et Cosmochimica Acta is the original reference for the oxygen isotope equilibration methodology, but the methods used today that involve flushing of sample vials with small quantities of carbon dioxide in otherwise pure helium are not available as details in these two publications of Epstein and Mayeda and Friedman and O'Neill, which were written well before the technology for continuous flow mass spectrometry was developed. Again, the Granskog et al 2011 paper provides a more accurate description of the methods used. I would start with that.

Page 5, second paragraph. It might be worthwhile to reference Smith et al. 2015 (http://dx.doi.org/10.1080/07011784.2014.9855120) either here and/or in the discussion, as they have added a significant amount of isotopic data for the Nelson River basin.

Page 5, line 17. I understand that a freshwater end-member isotopic value of -14‰ for Hudson Bay as a whole is used, but the more southern rivers such as the Nelson are clearly more enriched in heavy isotopes and this creates an inaccurate result for the sea ice melt contribution (Figure 2e). I am not sure this can be resolved easily but maybe in the future (with more extensive sampling) it would make more sense to divide Hudson Bay into areas influenced by northern rivers versus James Bay and the Nelson River and try to resolve more than one mixing process with runoff and melted sea ice.

Page 6, line 15-20. I am not sure why nutrient data, such as nitrate-phosphate ratios, and silicate are not also used to verify the presence of Pacific origin water; this would seem to be much more certain than TA, where multiple freshwater sources are present, although the salinity of 33.1 shown on Figure 3a seems convincing enough for the upper Arctic halocline that is characteristic of Pacific-origin water.

---

## Short Comment (SC1) · 5 Jul 2016

Dear Reviewers,

Many thanks for your helpful reviews. I have responded to all of your suggestions, and am now preparing to send the revised manuscript to my co-authors. Your suggestions have certainly improved the manuscript, and for that I am grateful.

Kind regards, William Burt

---

## Author Comment (AC1) · 18 Jul 2016

Attached are the detailed responses, on behalf of all authors, to the two reviews of the manuscript. We greatly appreciate the detailed comments and suggestions provided by these reviewers, and we feel that the changes made have led to significant improvements in all elements of the revised manuscript (i.e. the text, tables and figures). As a result, we have added a sentence to the acknowledgements section to personally thank the reviewers for their efforts.

Please also note the supplement to this comment: http://www.biogeosciences-discuss.net/bg-2016-197/bg-2016-197-AC1-supplement.pdf

[Figure]

**Supplement:**

**Authors Responses**

*Reviewer Comments are in blue
**Authors responses are in black

The authors feel that the comments made below have improved the manuscript substantially and have added a sentence to the acknowledgements as a token of our appreciation to both reviewers.

**Reviewer #1, Anonymous:**

**General comments:**

This work presents a nice data set from a region where few data has been collected, especially when focusing on the inorganic carbon cycle and how this is impacted by input from land. It is largely well
written and the figures illustrate the results. However there are some minor issues that the authors should consider before publication.

The authors are grateful for this detailed review. Various sections of the paper now have improved wording, and various statements have been added to the discussion section that improve the overall
interpretation of the data.

**Specific comments:**

One is an aspect that often is sees in the literature. The sentence that start in line 25 of page 1 reads that the
Arctic Ocean is particularly vulnerable to OA because of . . . low buffer capacity in cold waters. . . The low buffer capacity means that DIC is not changing much for a given change in pCO2, which also means that the concentration of CO32- does not change as much as it will in warmer water of less equilibrium DIC concentration. Thus the saturation state of calcium carbonate changes less in cold water compared to warm water for a given increase in pCO2. So even if the saturation state is lower to begin with does this mean
that these waters are more vulnerable? It is best to be more specific when making this kind of statement.

We appreciate this comment and understand the confusion created by the wording of this sentence. In the revised manuscript (starting line 263), we have clarified between solubility, and the change in solubility. Specifically, we have removed the parts of this sentence that refer to $CO_2$ solubility and buffering capacity,
and instead have described how changes in pH and carbonate ion concentration due to $CO_2$ uptake differ in cold, low alkalinity waters compared to warm, high alkalinity waters. We have also added a reference to Shadwick et al., 2013, wherein these concepts are discussed in detail.

A principle issue is that one analyse seawater but determine (or measure) TA, DIC or any other constituent. When analysing TA it means that one determines what TA consists of. One can claim that this is minor thing and that the word "analyse" often is used in this way, but that is not an excuse for not being correct.

The authors appreciate this point. The word 'analysis' is used incorrectly (and overused) throughout the manuscript. The authors have now used the terms 'measure' and 'determine' in the appropriate places.

Page 6, line 22. It is suggested that excess TA and salinity is added by brine, and on line 36 that sea ice
melt result in a negative deviation of TA. This is highly likely considering the data and other observations from the Fox basin, but can one thus conclude that this result show that no ikaite (calcium carbonate) is formed and trapped in the sea ice. I would urge the authors to expand on this topic.

The authors had not considered dissolution of Ikaite as a potential source of TA, so we thank the reviewer
for bringing this to our attention. A statement relating to this topic has been added (line 448), however, given the limitations of the dataset, we believe that any further discussion on this topic would be rather speculative.

On page 7 the term "benthic carbonate input" is used. I assume that what is meant that metal carbonate
dissolves at the sediment surface and that the carbonate ion is added to the bottom water. If so add "ion" after carbonate.

We appreciate this clarification, and have added 'ion' where appropriate.

Next to last sentence of the first paragraph on page 8. It is extremely important to take samples for salinity determination from the same flask as where the chemical samples are collected in stratified waters. This cannot be stressed enough and this contribution again well illustrates this.

We thank the reviewer for their appreciation of this result.

The section starting on line 12 at page 9. It would be interesting if the authors discuss how the observed C-13 signal is impacted by either decay of marine produced organic matter or terrestrial produced organic matter. Can the data be used to distinguish these sources?

The reviewer raises an interesting point here, as this analysis assumes all organic matter is of marine origin. Consequently, authors have added a note of caution to the end of this paragraph. However, we have also noted that the turnover of organic matter in marine systems is relatively rapid, thus a terrestrial signature will be very difficult to detect when sampling within the bay. Overall, these points (starting on line 544) are an interesting addition to the discussion.

Next paragraph. Here I have difficulties to understand how bottom water with high TA and DIC can be mixed up into the surface to give the high omega values without also mixing up salinity. Is it not more likely that the dissolution of metal carbonates occurs in the river drainage basin and is transported out to the estuary? Is there information on the mineral compositions of the different drainage basins that can add to the observations?

The authors agree that it is important to state that the majority of the carbonate dissolution likely occurs within the drainage basin, rather than in the estuary itself. However, we cannot rule out dissolution within the estuary entirely, thus in the revised manuscript we now state that carbonate dissolution may occur within the drainage basin and within the estuary (see statements starting at lines 463 and 553)

**General comments:**

This is a short descriptive paper describing the results of sampling in 2010 for total alkalinity and inorganic carbon, including the isotopic composition of DIC in Hudson Bay, with the use of oxygen isotopes as tool to evaluate runoff and melted sea ice contributions. It follows related studies by Azetsu-Scott et al. 2014 and Granskog et al. 2011, among others in the same region, so it is of value both as a point in a time-series for possible changes in the inorganic carbon system and to fill in the geographical gap for arctic data coverage within Hudson Bay. Because of the limited one time nature of the sampling, it is not clear whether changes can be readily determined, but the scope of the paper seems appropriate for the descriptive nature of the data set presented. I recommend a few improvements and pose some questions below for the authors to consider.

The authors greatly appreciate the thorough and thoughtful comments made in this review. Dr. Cooper clearly understands the scope of the paper, and his comments have led to significant improvements in all areas (text, figures, and tables) of the revised version.

**Specific comments:**

Figure 1. The map could be improved. It is plotted in a Mercator projection, which is not a polar projection, so distances are distorted, and it is not clear where the latitude and longitude markers are. The shaded areas denoting sea ice are also rather awkwardly illustrated and hard to see, such as the northern straits connecting to Hudson Bay. I think this could be easily re-done in a standard GIS program in a more appropriate polar map projection with better labeling. The map figure in the Granskog et al. 2011 paper that is referenced in the text, while still plotted in Mercator projection, is a much more attractive example of how to display the geographical information that is important to the data presentation.

The authors appreciate these comments, and have redone Figure 1. An orthogonal polar projection is now used, providing a less distorted map. The coloring has been altered to make the figure and labels more clear, and the scale of the map has been changed to avoid the northern section, which was not necessary.

Figure 2, and 6c. Why isn't the δ13C-DIC data coverage more complete? Again, as with Figure 1, the quality of these figures is only fair, and it could be improved by exporting out of Ocean Data View into a more robust graphics program, with appropriate labeling undertaken such as of the Nelson River (simply outlined in blue now). Also one of the software license requirements of Ocean Data View is to cite the software developers. "ODV may be used free of charge for non-commercial, non-military research and teaching purposes. If used in scientific papers, reports or posters, ODV must be referenced as follows: Schlitzer, R., Ocean Data View, odv.awi.de, 2015." Please add to the references or acknowledgements.

δ13C-DIC was only sampled during the first part of the cruise. For clarity, this has been stated explicitly in

160 the captions of Figures 2 and 6.

As for figure 2, the authors feel that these ODV plots adequately illustrate the key discussion points in the manuscript. Steps were taken to try and improve the quality of the image, such as labeling of the Nelson River, but it was decided to leave this as a descriptive label in the caption in order to maintain a larger text size within the panels.

165 Finally, the authors appreciate pointing out the ODV license requirement, and have added the reference to the acknowledgements.

Figure 5 caption. The discrepancy with rosette bottle salinities can be caused by inadequate flushing time
170 before bottles are closed at depth. Is the flushing time used by the CTD operator known?

The flushing time at each depth was approximately 1 minute, this has been confirmed by the CTD operator. The authors appreciate that this is useful information for the reader, and thus we have added this statement to the methods section (line 324), as well as to the caption in Figure 5.

Table 1. The one-time sampling of these rivers is a limitation for this type of study that depends upon integrated and flux weighted data. The table could be expanded and include references to any other available data for these rivers.

The authors agree and appreciate this recommendation. Table 1 has been expanded to include discharge and dissolved organic carbon (both of which were discussed in the manuscript but not shown in the table), as well as prior 18O measurements. The required references have also been added to the table.

Page 4, line 25. Friedman and O'Neill (not Friedman et al. 1977) is really not a very appropriate reference for this method of determining oxygen isotopes. It describes the isotopic fractionation factors associated with equilibration of water with carbon dioxide, but it is not really a description of the method. Epstein and Mayeda, 1953 in Geochemica et Cosmochimica Acta is the original reference for the oxygen isotope
190 equilibration methodology, but the methods used today that involve flushing of sample vials with small quantities of carbon dioxide in otherwise pure helium are not available as details in these two publications of Epstein and Mayeda and Friedman and O'Neill, which were written well before the technology for continuous flow mass spectrometry was developed. Again, the Granskog et al 2011 paper provides a more accurate description of the methods used. I would start with that.

The authors tend to agree that the citation to the Friedman and O'Neil paper is outdated, and frankly, unnecessary. We have removed this citation and, instead, have finished describing the method to an adequate degree so as to not require any citation (see paragraph starting at line 361).
.

Page 5, second paragraph. It might be worthwhile to reference Smith et al. 2015 (http://dx.doi.org/10.1080/07011784.2014.9855120) either here and/or in the discussion, as they have added a significant amount of isotopic data for the Nelson River basin.

We appreciate bringing this paper to our attention. Data from Smith et al. 2015 is now presented in Table 1, and is referenced in the revised manuscript (line 477)

Page 5, line 17. I understand that a freshwater end-member isotopic value of -14‰ for Hudson Bay as a whole is used, but the more southern rivers such as the Nelson are clearly more enriched in heavy isotopes and this creates an inaccurate result for the sea ice melt contribution (Figure 2e). I am not sure this can be resolved easily but maybe in the future (with more extensive sampling) it would make more sense to divide Hudson Bay into areas influenced by northern rivers versus James Bay and the Nelson River and try to resolve more than one mixing process with runoff and melted sea ice.

The authors agree that this oversimplification is an issue, and multiple regional endmembers were discussed during manuscript preparation. However, given the limitations of this 'snapshot' dataset and, consequently, the limited scope of this particular section, we believe the simplified method is justifiable here. We do, however, fully understand the necessity to resolve this issue, and have added a sentence to this effect in the conclusion of the revised manuscript (line 583).

Page 6, line 15-20. I am not sure why nutrient data, such as nitrate-phosphate ratios, and silicate are not also used to verify the presence of Pacific origin water; this would seem to be much more certain than TA, where multiple freshwater sources are present, although the salinity of 33.1 shown on Figure 3a seems convincing enough for the upper Arctic halocline that is characteristic of Pacific-origin water.

The authors do agree that a comparison between alkalinity and nutrients as watermass tracers in the Hudson Bay system would be of interest. However, there is insufficient nutrient data available from this cruise to undergo this comparison, especially given that this is a very complex discussion that may be more suitable as a separate study altogether. For example, nitrate and phosphate are not always ideal watermass tracers due to their relatively non-conservative behaviour compared to the more conservative alkalinity. Also, in shallow systems like the Canadian Archipelago, processes such as denitrification can cause major complications in a nutrient-based watermass analysis.

[revised manuscript text omitted]